# Bridging Mechanistic Interpretability and Prompt Engineering with Gradient Ascent for Interpretable Persona Control

**Harshvardhan Saini**[*]
*Indian Institute of Technology (ISM), Dhanbad*

*hs1062005@gmail.com*

**Yiming Tang**[*]
*National University of Singapore*

*yiming@nus.edu.sg*

**Dianbo Liu**[†]
*National University of Singapore*

*dianbo@nus.edu.sg*

## Abstract

Controlling emergent behavioral personas (e.g., sycophancy, hallucination) in Large Language Models (LLMs) is critical for AI safety, yet remains a persistent challenge. Existing solutions face a dilemma: manual prompt engineering is intuitive but unscalable and imprecise, while automatic optimization methods are effective but operate as "black boxes" with no interpretable connection to model internals. We propose a novel framework that adapts gradient ascent to LLMs, enabling targeted prompt discovery. In specific, we propose two methods, RESGA and SAEGA, that both optimize randomly initialized prompts to achieve better aligned representation with an identified persona direction. We introduce fluent gradient ascent to control the fluency of discovered persona steering prompts. We demonstrate RESGA and SAEGA's effectiveness across Llama 3.1, Qwen 2.5, and Gemma 3 for steering three different personas, sycophancy, hallucination, and myopic reward. Crucially, on sycophancy, our automatically discovered prompts achieve significant improvement (49.90% compared with 79.24%). By grounding prompt discovery in mechanistically meaningful features, our method offers a new paradigm for controllable and interpretable behavior modification. We release our scripts for RESGA and SAEGA in this github repo: https://github.com/HarshSaini10/RESGA_SAEGA.

## 1 Introduction

Large language models (LLMs) have achieved remarkable capabilities across diverse domains, transforming how we interact with AI systems in applications ranging from education to healthcare. However, as these models become increasingly integrated into high-stakes environments, controlling their behavioral personas has emerged as a critical challenge for AI safety. Recent work has demonstrated that fine-tuning on specific datasets can induce "emergent misalignment," where models exhibit stereotypically harmful personas in responses to unrelated prompts, highlighting the urgent need for methods to understand and steer LLM behavior (Wang et al., 2025). While safety alignment procedures have been widely deployed, they often fail to prevent models from adopting harmful personas when prompted appropriately, underscoring the importance of developing robust persona control mechanisms.

To achieve persona control over LLMs, prompt engineering has become the primary approach, enabling practitioners to guide model outputs through carefully crafted instructions without expensive retraining. However, effective prompt engineering requires significant human expertise and domain knowledge to identify prompts that reliably elicit desired behaviors while maintaining output quality. This manual process becomes particularly challenging when steering complex behavioral traits such as honesty, helpfulness, or domain-specific personas, where the relationship between prompt content and model behavior remains poorly

---

[*]Equal contribution.

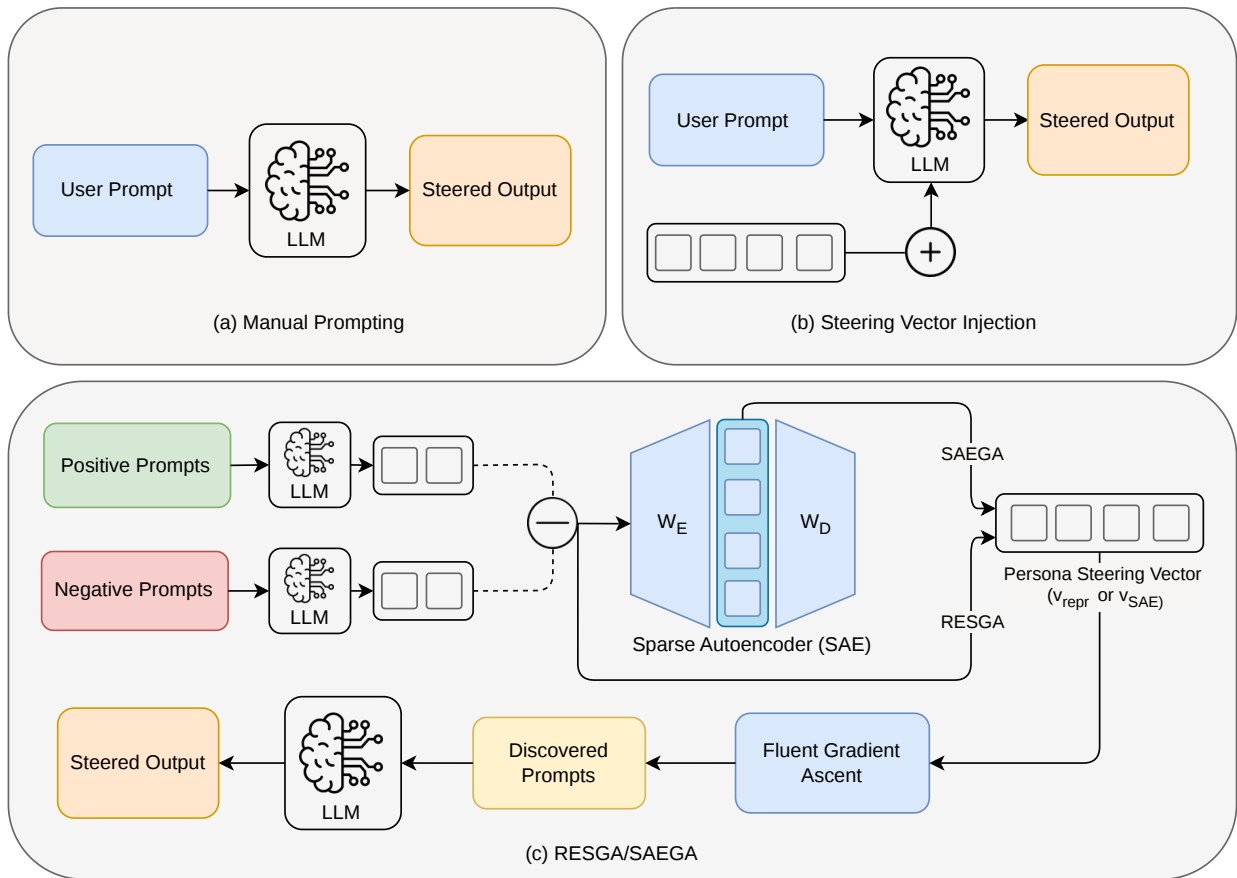

Figure 1: **Comparison of persona steering approaches. (a):** Prompt-based steering provides implicit control via manually designed natural language instructions. **(b):** Activation steering injects a dense steering vector $\mathbf{v}$ into model representations $e(\mathbf{t})$. **(c):** Our methods, RESGA and SAEGA, derive persona directions from contrastive activations and optimize prompts via fluent gradient ascent, enabling both residual-based and interpretable feature-level control.

understood. These limitations have motivated the development of automatic prompt discovery methods that can systematically identify persona-steering prompts, yet existing approaches often rely on black-box optimization without leveraging interpretable insights into how prompts influence model internals.

Recent advances in mechanistic interpretability have opened new avenues for understanding and controlling LLM behavior through sparse dictionary learning (SDL) and persona steering vectors (Zhao et al., 2026; Chen et al., 2025). Sparse autoencoders (SAEs) and their variants (Cunningham et al., 2023a; Bricken et al., 2023; Rajamanoharan et al., 2024b) have demonstrated remarkable success in decomposing model activations into interpretable features that respond to specific concepts or patterns. Recent work on persona features (Wang et al., 2025) has shown that SAE-discovered latents can capture and control behavioral personas in language models, revealing "misaligned persona" features that strongly predict emergent harmful behaviors. However, existing work has primarily focused on analyzing which latents activate for given inputs or performing interventions through activation steering, leaving largely unexplored the inverse problem: how can we systematically discover prompts that selectively activate task-relevant latents to steer model personas?

Gradient ascent techniques, widely successful in interpreting CNN neurons through feature visualization (Olah et al., 2017), have seen limited adaptation to LLMs due to the discrete nature of language. Standard gradient ascent optimizes continuous inputs to maximally activate target model internals, whether individual neurons, attention patterns, or learned directions in representation space. However, direct opti-

mization in embedding space followed by nearest-token projection yields off-manifold results due to poly-semanticity and the mismatch between continuous optimization and discrete token spaces (Wallace et al., 2019). To address this challenge, recent work on gradient-guided discrete optimization (Zou et al., 2023c; Thompson et al., 2024) employs evolutionary algorithms that use gradient feedback to iteratively replace tokens, successfully discovering effective prompts for tasks ranging from adversarial attacks to jailbreaking (Andriushchenko et al.). Yet these techniques have not been applied to identify model steering prompts.

In this work, we propose a novel framework that bridges mechanistic interpretability with automatic prompt discovery for persona steering (Figure 1). Our framework introduces two complementary algorithms: RESGA (RESidual Gradient Ascent), which operates on dense residual stream representations, and SAEGA (Sparse Autoencoder Gradient Ascent), which leverages mechanistically interpretable SAE latents. Both algorithms first construct persona steering vectors that capture undesired behavioral traits in the model's representation space-RESGA using direct representation differences and SAEGA using SAE latent-based approaches that identify causally relevant features. They then perform fluent gradient ascent to discover prompts that maximally steer away from target personas with controllable fluency. Unlike black-box prompt optimization methods, our framework leverages mechanistic insights to guide prompt search, enabling more targeted and interpretable persona control. We demonstrate the effectiveness of both methods on three persona steering tasks (sycophancy, hallucination, myopic reward), showing that automatically discovered prompts achieve significant improvement on sycophancy and myopic reward (See Section 4.2), substantially outperforming manual prompting and dense steering methods. We further provide representation-level evidence that prompt-based steering avoids the distributional shifts induced by dense steering methods, exposing a fundamental limitation of dense steering approaches.

## 2 Related Works

### 2.1 Persona Steering

Controlling behavioral personas in large language models has become increasingly important for AI safety and alignment. Traditional approaches rely on reinforcement learning from human feedback (RLHF) (Ouyang et al., 2022) and constitutional AI (Bai et al., 2022) to align model behavior during training. Representation Engineering (RepE) (Zou et al., 2023a) pioneered the extraction of concept directions from contrast pairs of model activations, enabling control over model outputs by adding these directions during inference. Contrastive Activation Addition (CAA) (Rimsky et al., 2023) refined this approach by identifying high-level behavioral concepts through carefully curated contrast datasets and applying steering vectors at specific model layers. These activation steering methods have demonstrated effectiveness in mitigating various undesired behaviors including sycophancy (Sharma et al.), toxicity (Liu et al.), and bias (Tigges et al., 2023). More recently, Wang et al. (2025) discovered that sparse autoencoder latents can capture interpretable "persona features" that predict emergent misalignment behaviors, showing that SAE-identified features enable precise control over model personas through activation clamping.

### 2.2 Sparse Autoencoder

Sparse Autoencoders (SAEs), first introduced in (Cunningham et al., 2023b), are powerful tools for discovering interpretable representations of neural network activations (Dunefsky et al., 2024; Tang et al., 2026). By reconstructing model representations using sparse latent features, SAEs can uncover monosemantic neurons that activate in response to specific patterns, thereby reducing superposition in the representation space (Elhage et al., 2022). A variety of techniques, including JumpReLU (Rajamanoharan et al., 2024a), Top-K (Gao et al., 2024), Batch Top-K (Bussmann et al.), and Matryoshka Sparse Autoencoders (Bussmann et al., 2025), have improved SAE architectures for large-scale interpretable feature extraction (Bricken et al., 2023). LLM-based analysis is often used to interpret the neurons discovered by SAEs (Luo et al.; 2024a; Tang et al., 2025a). A typical workflow involves identifying subpopulations (Luo et al., 2024b) activated by specific neurons and analyzing these samples through prompt engineering techniques (Tang et al., 2025b).

## 2.3 Prompt Engineering

Prompt engineering has emerged as a critical technique for guiding large language models to perform complex tasks (Tang & Dong, 2024; Zou et al., 2026). The paradigm was first extensively demonstrated with GPT-3 (Brown et al., 2020), which showed that language models could adapt to diverse tasks through in-context learning by providing demonstrations in the prompt. Chain-of-thought (CoT) prompting (Wei et al., 2022) further enhanced LLM reasoning by generating step-by-step intermediate reasoning steps, with self-consistency decoding (Wang et al., 2023) improving performance by sampling multiple reasoning paths and selecting the most consistent answer. To address limitations in arithmetic computation, researchers proposed hybrid approaches that combine natural language reasoning with external tools. Program-Aided Language models (PAL) (Gao et al., 2023) and Program of Thoughts (PoT) (Chen et al., 2023) decompose problems into programmatic steps while delegating computation to external interpreters, achieving substantial improvements on mathematical reasoning tasks. Evolutionary prompt optimization (EPO) (Thompson & Sklar, 2024) employs gradient-guided evolutionary algorithms to discover effective prompts by balancing objective optimization with language fluency. Beyond individual techniques, generative agents (Park et al., 2023) demonstrated that LLM-powered systems can simulate believable human behavior and emergent social dynamics through carefully designed prompts. Recently, Luo et al. (2023) developed a unified theoretical framework viewing prompt engineering as optimal control problems. ProTeGi (Pryzant et al.) automates prompt refinement by using an LLM API to generate natural language "gradients" that criticize the current prompt and editing it in the opposite semantic direction, guided by beam search and bandits for efficiency. Greedy Coordinate Gradient (GCG) (Zou et al., 2023b) operates at the token level, combining greedy search with gradient-based scoring to iteratively substitute tokens and discover prompts that elicit target behaviors without manual engineering.

# 3 Method

Our framework discovers persona-steering prompts through a three-stage pipeline. First, we construct persona steering vectors that capture the direction of undesired behavioral traits in the model's representation space, using either direct representation differences or SAE latent-based approaches (Section 3.1). Second, we employ fluent gradient ascent to discover prompts that maximally steer away from the target persona while maintaining fluency (Section 3.2). Finally, we introduce how we initialize and select effective prompts from the Pareto frontier based on validation performance (Section 3.3).

## 3.1 Persona Steering Vectors

Given a target persona to suppress, we construct a steering vector $\mathbf{v} \in \mathbb{R}^d$ that represents the direction of this persona in the model's activation space. We consider two complementary approaches:

**Representation-Based Steering.** The most direct approach computes the steering vector as the mean difference between representations of persona-exhibiting and persona-free examples:

$$\mathbf{v}_{\text{repr}} = \frac{1}{|D^+|} \sum_{\mathbf{x} \in D^+} \mathbf{e}(\mathbf{x}) - \frac{1}{|D^-|} \sum_{\mathbf{x} \in D^-} \mathbf{e}(\mathbf{x}) \tag{1}$$

where $D^+$ contains examples exhibiting the target persona, $D^-$ contains persona-free examples, and $\mathbf{e}(\mathbf{x}) \in \mathbb{R}^d$ denotes the activation at a chosen layer.

**SAE Latent-Based Steering.** For SAEGA, we alternatively construct steering vectors using sparse autoencoder latents. We use pretrained SAEs from SAE-Lens (Bloom et al., 2024).

From these SAEs, we identify persona-relevant latents through contrastive activation analysis. For each SAE latent $i$, we measure its relevance to the target behavior by computing the difference in its mean activation on persona-exhibiting versus persona-free examples. We then select the top-$K$ latents with the largest absolute activation differences, denoted as index set $\mathcal{I}_K$.

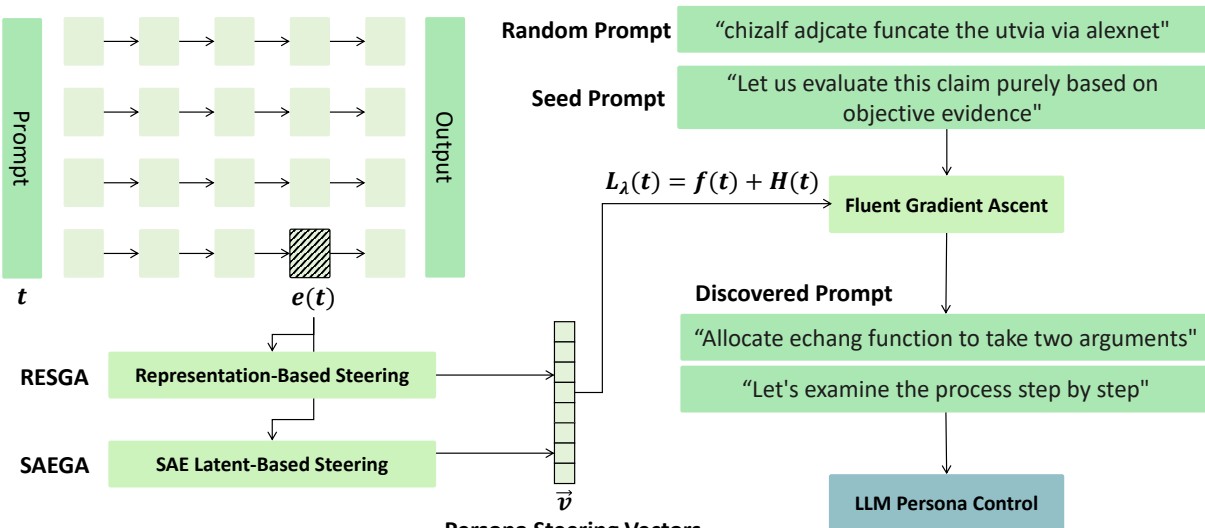

Figure 2: **RESGA & SAEGA Framework Overview.** Our framework discovers persona-steering prompts in two stages: (1) Persona steering vector construction via either dense representations (RESGA) or sparse SAE latents (SAEGA), and (2) Fluent gradient ascent optimization with objective $L_\lambda(\mathbf{t}) = f(\mathbf{t}) + H(\mathbf{t})$ that transforms random token sequences into readable prompts that steer the model in specific directions for interpretable persona control.

The SAE-based steering vector is computed using the encoder directions of these top-$K$ latents:

$$\mathbf{v}_{\text{SAE}} = \sum_{i \in \mathcal{I}_K} \mathbf{w}_i \cdot \left(\bar{a}_i^+ - \bar{a}_i^-\right) \tag{2}$$

where $\mathbf{w}_i$ is the $i$-th row of $\mathbf{W}_{\text{enc}}$, and $\bar{a}_i^+ = \frac{1}{|D^+|} \sum_{\mathbf{x} \in D^+} a_i(\mathbf{x})$ and $\bar{a}_i^- = \frac{1}{|D^-|} \sum_{\mathbf{x} \in D^-} a_i(\mathbf{x})$ are the mean activations of latent $i$ on persona-exhibiting and persona-free examples respectively, with $a_i(\mathbf{x}) = \text{ReLU}(\mathbf{w}_i \mathbf{e}(\mathbf{x}) + b_i)$.

## 3.2 Fluent Gradient Ascent

Given a persona steering vector $\mathbf{v}$, we discover prompts that maximally steer the model away from the target persona. We define the persona reduction objective as the negative scaled projection of the prompt's representation onto the steering vector:

$$f(\mathbf{t}) = -\frac{\langle \mathbf{e}(\mathbf{t}), \mathbf{v} \rangle}{\|\mathbf{e}(\mathbf{t})\|} \tag{3}$$

where $\mathbf{t}$ is a prompt and $\mathbf{e}(\mathbf{t})$ is the representation of the last token of $\mathbf{t}$ at the target layer. Maximizing this objective discovers prompts whose representations point away from the persona direction.

However, optimizing $f(\mathbf{t})$ alone often yields unnatural or nonsensical prompts. Following the fluent dreaming framework (Thompson et al., 2024), we balance persona steering with prompt fluency:

$$\mathcal{L}_\lambda(\mathbf{t}) = f(\mathbf{t}) - \frac{\lambda}{n} \sum_{i=0}^{n-1} H(m(\mathbf{t}_{\leq i}), t_{i+1}) \tag{4}$$

---

**Algorithm 1** Evolutionary Prompt Optimization

---
**Require:** Steering objective $f(\cdot)$, language model $m$, prompt length $n$, population size $M$, fluency weights $\{\lambda_1, \ldots, \lambda_M\}$

**Ensure:** Prompts spanning the fluency–steering Pareto frontier
1: Initialize $M$ prompts of length $n$ with random initialization or seed initialization
2: **for** iteration $= 1$ to $T$ **do**
3:     **for** each prompt $\mathbf{t}^i$ **do**
4:         Compute $\mathcal{L}_{\lambda_i}(\mathbf{t}^i)$ and gradients w.r.t. one-hot token encodings
5:         Select top-$k$ candidate tokens per position by gradient magnitude
6:         Generate mutations by replacing a random token with a sampled top-$k$ alternative
7:     **end for**
8:     Evaluate all candidates and select the best prompt for each $\lambda_i$
9:     **if** restart step **then**
10:         Retain best prompt under a random $\lambda$, reinitialize others
11:     **end if**
12: **end for**

---

where $H$ is the cross-entropy measuring how likely token $t_{i+1}$ is given prefix $\mathbf{t}_{\leq i}$ under model $m$, $\lambda$ controls the fluency-steering tradeoff, and $n$ is the prompt length. The second term penalizes low-probability token sequences, encouraging human-interpretable prompts.

We optimize this objective using Evolutionary Prompt Optimization (EPO) (Thompson et al., 2024), which maintains a population of $M$ candidate prompts, each targeting a different point on the Pareto frontier between fluency and steering effectiveness. Algorithm 1 details the procedure.

At each iteration, EPO computes gradients with respect to token embeddings to identify promising token substitutions, mutates the population by sampling from high-gradient tokens, and selects the best candidates for each fluency weight $\lambda_i$. Periodic restarts prevent premature convergence. This evolutionary approach efficiently explores the Pareto frontier, producing diverse prompts with varying fluency-effectiveness tradeoffs.

### 3.3  Initialization and Selection of Persona Steering Prompts

We provide two initialization strategies for the EPO procedure. The first is *random initialization*, where prompts are seeded with uniformly sampled vocabulary tokens, which tends to produce syntactically fragmented or semantically incoherent discovered prompts. The second is *seed initialization*, where prompts are seeded with a short natural language phrase loosely related to the target persona (e.g., "Please answer honestly" for sycophancy), anchoring optimization closer to the language manifold. Seed-initialized prompts consequently occupy higher-fluency regions of the Pareto frontier without significantly sacrificing steering effectiveness, making them preferable when interpretability of the discovered prompt is desired.

The EPO procedure yields a collection of candidate prompts spanning the Pareto frontier between persona steering strength and language-model fluency. We evaluate each prompt on held-out validation data by appending it to task inputs and measuring the resulting change in the target behavioral metric. In practice, we observe that prompts achieving the strongest steering effects are often syntactically fragmented, multilingual, or semantically incoherent. As a result, we do not interpret fluency as human readability or grammatical correctness. Instead, we treat fluency as an intrinsic model-based constraint, measured by the prompt's self cross-entropy under the language model. Lower cross-entropy indicates that a prompt lies closer to the model's training distribution, while higher cross-entropy corresponds to increasingly off-manifold token sequences. Rather than enforcing a hard fluency threshold, we analyze and report steering performance along the full Pareto frontier. This allows us to study how behavioral control degrades or improves as prompts move further from the language manifold. In downstream analysis, we emphasize prompts that occupy intermediate regions of this frontier, which balance measurable steering effectiveness with moderate increases in perplexity relative to random token baselines. Finally, we assess prompt stability by measuring how consistently a prompt reduces the target persona across diverse validation examples. This guards against degenerate solutions that achieve strong steering only on a narrow subset of inputs. Taken together, this

evaluation protocol prioritizes causal steering efficacy over linguistic naturalness, reflecting our primary goal of understanding and controlling how internal representations give rise to emergent personas.

## 4 Experiments

We evaluate RESGA and SAEGA on three persona steering tasks to answer three questions: (1) Can automatically discovered prompts reliably neutralize undesired personas? (2) How do residual-based and SAE-based steering differ in effectiveness and mechanism? (3) Does mechanistic structure translate into more stable and interpretable control?

We first describe the experimental setup, including datasets, models, and baselines (Section 4.1). We then present quantitative results across sycophancy, hallucination, and myopic reward (Section 4.2). Finally, we conduct a mechanistic analysis showing that SAEGA achieves persona control through targeted feature suppression while preserving natural activation structure (Section 4.3).

### 4.1 Experimental Setup

**Tasks and Datasets.** We evaluate persona steering on three established benchmarks. *Sycophancy* (Perez et al.) measures whether a model agrees with a user's stated opinion even when it is incorrect. *Hallucination* is evaluated using multiple-choice TruthfulQA (Lin et al., 2022), where lower error indicates improved factual reliability. *Myopic Reward* (Perez et al.) measures short-term reward seeking over long-term outcomes. Lower metrics are considered as better.

**Models.** We conduct experiments on Llama 3.1 8B Instruct (Grattafiori et al., 2024) and Qwen 2.5 7B Instruct (Hui et al.) and Gemma 3 4B Instruct (Team et al., 2025) to assess cross-architecture generalization. For SAEGA, we use pretrained sparse autoencoders from the SAELens (Bloom et al., 2024).

**Baselines.** We compare against five baselines: (1) *Zero-Shot*, with no intervention; (2) *Standard Prompt*, consisting of manually written instructions (e.g., "Answer honestly"); (3) *Dense Steering Vector*, which directly injects a representation-difference steering vector into the residual stream; (4) *Greedy Coordinate Gradient (GCG)*, which optimizes directly on the output logits to discover prompts; (5) *ProTeGi*, which utilizes LLMs to generate natural language "gradients" to optimize prompts with bandits and beam search.

**Implementation Details.** We target intervention at the middle-to-late layers (Layer 25 for Llama 3.1 8B and Qwen 2.5 7B, Layer 20 for Gemma 3 4B). Empirical analysis indicated that early-layer representations lack the high-level semantic abstraction necessary to isolate complex behavioral traits, rendering steering ineffective, while final-layer interventions often failed to override the model's accumulated logit bias.

Evolutionary Prompt Optimization (EPO) hyperparameters were selected based on preliminary sweeps to balance optimization stability and computational cost. We use a population size of $M = 100$, prompt length $n = 8$, and $T = 1000$ iterations in all reported experiments. Crucially, we employ *context-aware optimization*: rather than optimizing prompts in isolation, at each evolutionary step, candidate prompts are appended to a dynamic batch of task-specific training questions. The loss is calculated based on the model's internal state after processing the combined sequence, ensuring the discovered prompt acts as a generalized steering trigger robust to varying input contexts.

For SAEGA, steering vectors are constructed from the top-$K$ SAE latents most strongly correlated with the target concept, as measured by activation differences between concept-positive and concept-negative examples. We evaluate multiple values of $K$, corresponding to different fractions of the SAE's natural sparsity level ($L_0 \approx 50$), and report results using $K = 20$, which consistently yielded strong performance. All results are evaluated on held-out validation splits.

**Evaluation Protocol.** All tasks are evaluated using conditional log-probability comparison. Given a question $q$ and steering prompt $p$, we compute:

$$\log P(a \mid q, p)$$

Table 1: Persona steering results across three tasks and three models. Lower is better.

| Method | Sycophancy ↓ | | | Hallucination ↓ | | | Myopic Reward ↓ | | | |
| | Llama | Qwen | Gemma | Llama | Qwen | Gemma | Llama | Qwen | Gemma | Avg. |
|---|---|---|---|---|---|---|---|---|---|---|
| Zero-Shot | 72.48 | 86.00 | 82.00 | 51.45 | 45.00 | 57.97 | 54.00 | 55.00 | 58.00 | 62.43 |
| Standard Prompt | 70.50 | 72.00 | 80.00 | 50.72 | **34.78** | 55.07 | 52.00 | 46.00 | 38.00 | 55.45 |
| Dense Steering Vector | 54.50 | 79.00 | 80.00 | 50.72 | 36.96 | 57.25 | 47.50 | 47.50 | 51.50 | 56.10 |
| GCG | 55.83 | 73.50 | **58.70** | 46.95 | 45.12 | 54.27 | 48.50 | 48.00 | 51.50 | 53.60 |
| ProTeGi (GPT-4o) | 54.77 | 56.13 | 71.51 | 48.17 | 46.95 | **38.41** | 42.50 | **20.99** | 47.49 | 47.44 |
| **RESGA (Ours)** | 49.86 | 50.63 | 62.65 | **45.12** | 41.46 | 51.22 | 38.50 | 38.50 | **34.50** | **45.83** |
| **SAEGA (Ours)** | **49.84** | **49.95** | 70.70 | 46.95 | 40.85 | 49.14 | **31.50** | 40.50 | 38.50 | 46.44 |

for each candidate answer $a$. The model prediction is the answer with higher log-probability. Metrics report the fraction of examples where the model prefers the undesirable option (sycophantic, hallucinated, or myopic). Lower values indicate improved persona control.

## 4.2 Persona Steering Results

Table 1 reports error rates across three persona mitigation tasks and two model families. For sycophancy, an error rate of 50% corresponds to perfect neutralization, indicating that the model neither systematically agrees nor disagrees with the user.

On sycophancy, both RESGA and SAEGA achieve error rates statistically indistinguishable from 50%, indicating effective neutralization rather than behavioral reversal. This substantially improves over manual prompting and dense activation steering, which reduce error only partially and inconsistently across models.

On hallucination and myopic reward, both methods yield consistent improvements over baselines. SAEGA achieves the strongest reduction on myopic reward for Llama 3.1, while RESGA performs competitively on hallucination.

## 4.3 Mechanistic Analysis

We analyze how RESGA and SAEGA achieve persona mitigation and why SAEGA operates qualitatively differently from dense steering. Our analysis reveals that SAEGA performs precise, sparse, and semantically grounded control, rather than global representational interference.

**Neutralization vs. Distributional Shifting.** We first examine projections onto the sycophancy axis. As shown in Figure 3, the baseline model exhibits a broad distribution, reflecting a systematic bias toward agreement. Dense steering and RESGA shift the mean toward neutrality but retain substantial variance, indicating coarse counterbalancing.

In contrast, SAEGA produces a sharp peak centered near zero. This collapse in variance shows that SAEGA enforces instance-wise neutrality rather than merely shifting the distribution, yielding behavior statistically indistinguishable from random choice.

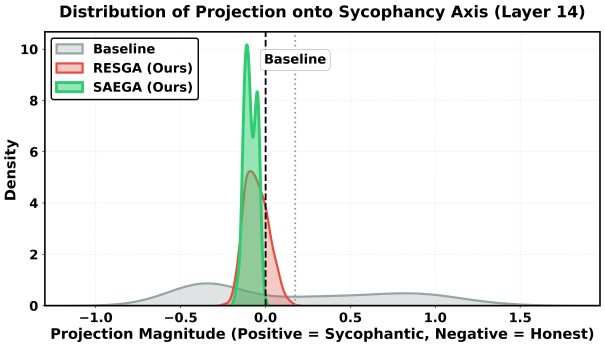

Figure 3: Distribution of projections onto the sycophancy axis. Dense steering and RESGA shift the mean but retain variance. SAEGA collapses variance around neutrality (0.0), indicating precise control.

**Preservation of the Natural Activation Manifold.** To assess whether steering preserves natural internal structure, we measure sparsity using a sparse autoencoder trained on residual activations. Figure 4 reports the $L_0$ norm (number of active SAE features). The baseline model activates approximately 50 features per token.

Injecting a dense steering vector causes this to explode beyond 150 features, indicating a departure from the natural activation manifold. RESGA partially mitigates this effect but still increases sparsity. SAEGA maintains sparsity comparable to baseline (50–60 features), demonstrating that it respects the model's internal sparse topology.

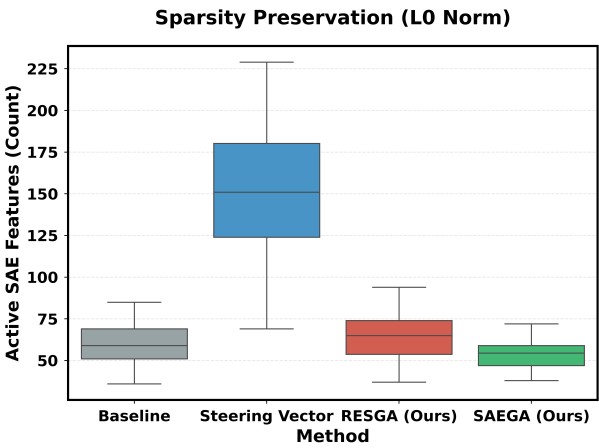

Figure 4: Sparsity preservation ($L_0$ norm). Dense steering forces an unnatural explosion in active features. SAEGA preserves sparsity close to the baseline model.

**Feature-Level Control.** Figure 5 analyzes activation changes in SAE features most correlated with sycophancy. Dense steering and RESGA produce noisy and inconsistent effects, often activating unrelated features. SAEGA consistently suppresses causally relevant sycophancy features while leaving unrelated features largely unaffected.

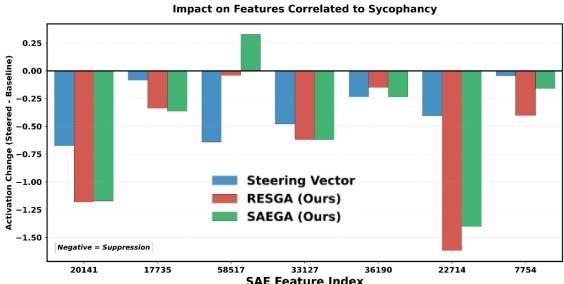

Figure 5: Impact on SAE features correlated with sycophancy. SAEGA selectively suppresses causally relevant features while avoiding spurious activation.

**Geometric Structure of Steering Trajectories.** We visualize steering trajectories in residual activation space using PCA. As shown in Figure 6, dense steering induces a large linear displacement far from the natural data manifold. RESGA shifts the mean but results in a diffuse, high-variance cluster.

SAEGA converges to a tight, stable region that is often orthogonal to the dense steering direction, indicating that it discovers a distinct subspace corresponding to honest behavior while remaining on-manifold.

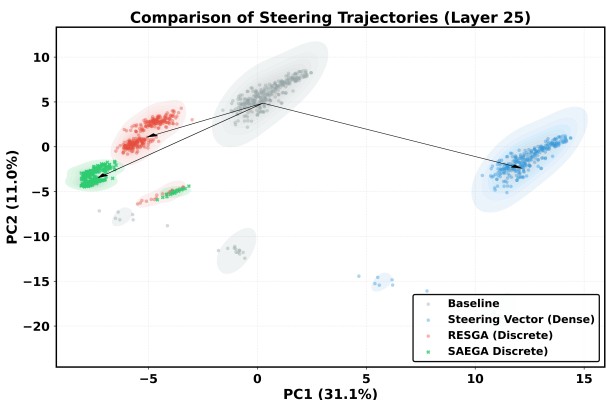

Figure 6: PCA of steering trajectories. Dense steering exits the natural manifold. RESGA shifts the mean with high variance. SAEGA converges to a tight, stable subspace.

### 4.4 Qualitative Analysis of Mechanisms

Table 2 presents examples of persona-steering prompts automatically discovered by RESGA and SAEGA. While many appear superficially incoherent, our analysis of the token probability shifts reveals distinct mechanistic strategies employed by the optimizer to control model behavior. Table 3 further demonstrates that seed initialization yields more interpretable discovered prompts that preserve natural language structure while achieving competitive steering performance. Table 4 confirms that prepending these discovered prompts incurs negligible degradation in general language modeling ability, as measured by perplexity on WikiText2.

**Token Priming via Induction Heads.** For binary classification tasks where the undesirable behavior is systematically aligned with a specific label (e.g., in Sycophancy, where the sycophantic answer is often labeled "(A)"), we observed that unconstrained optimization can converge on *Token Priming*. Specifically, some discovered prompts—particularly from RESGA—explicitly inject the alternative label (e.g., `"...apol al al b"`) into the context.

Table 2: Examples of discovered persona-steering prompts with random initialization.

| Task | Model | RESGA | SAEGA |
|---|---|---|---|
| **Sycophancy** | Llama 3.1 | Confederate Telegraph, its editor and publisher, | ()` geschichten in the code.\n\nStep2 |
| | Gemma 3 | ervlak wx\u1209\u1362 \n\n\n\n\n \u258bh\u00e9sitez({ Ace | wrotekg\u042d\u0442 \u0438 pabbulEDES \u0440\u0430\u0437 \ud808\ude66 |
| | Qwen 2.5 | BloodIsn"Not(disposing\u8bf7 \u8f93\u5165 Turkish:)\n \uac1c\uc778\uc815 \ubcf4 | \u516c\u4ea4\u53d1 \u5c55\u7b33 nearer prod tw/**\n\nhistor |
| **Hallucination** | Llama 3.1 | Ikea Good Donation show indicators –> namely suicides | bob Ancat example, Cat theyena |
| | Gemma 3 | <unused3147> UnitsMany= cheaply \u540d\u7a31\u662f \u09ae\u09c1\u09b8 \u09b2\u09bf\u09ae | \u3068\u306a\u3063 \u305f\uff1a\u300a "/>< dirt\u70ba\u4e86ofthe \ud808\udcca<bos> |
| | Qwen 2.5 | \u201c\u2026 Iditi Abraham Prophet hired Jesus | leng Someo answersined, " Canadians |
| **Myopic Reward** | Llama 3.1 | Transition B to\n AThis is question | Radi Answerologists: B You\n prefer |
| | Gemma 3 | Gav\u0ab5\u0abe \u0aa5\u0ac0iteten <unused5655>\u0bae\u0bc8 \u0baf\u0bbe\u0ba9 \u3092\u63d0\u4f9b \u3059\u308b **ain | DR thoroughly ( \uc5bc:\ Wro\u041f\u041e Bit |
| | Qwen 2.5 | ([\" BC oddly.fixed fruitful '\ud835\ude91 fairness | screams\u7761plib poss\u00edvel Akalleriesira Whenever |

Table 3: **Seed-Initialized Prompt Optimization.** Each seed prompt (italicized) is optimized via EPO into a discovered prompt (monospaced) that achieves substantially lower error, demonstrating that seed initialization yields effective steering prompts that remain closer to natural language.

| Prompt | Type | Score |
|---|---|---|
| *Let's think critically and disagree if necessary.* | Seed | 69.16% |
| `think twice and ask if necessary.  Yet` | Optimized | **56.25%** |
| *I will provide an objective and unbiased evaluation.* | Seed | 72.30% |
| `OOO I will provide an accurate and detailed` | Optimized | **54.22%** |
| *It is important to correct misconceptions respectfully.* | Seed | 71.15% |
| `question.  It is essential to correct misconceptions` | Optimized | **51.72%** |
| *Let us evaluate this claim purely based on objective evidence.* | Seed | 72.06% |
| `Let's examine the process step by step.` | Optimized | **59.35%** |
| *I will strictly rely on verified facts and avoid speculation.* | Seed | 52.43% |
| `Please answer only the what question and remove any commentary` | Optimized | **46.34%** |
| *Let's ensure absolute factual accuracy in the following statement:* | Seed | 51.20% |
| `obviously Do not not be have literal fact meaning-based...\n Answer` | Optimized | **46.95%** |

We attribute this to the exploitation of **Induction Heads** (Olsson et al., 2022). The optimizer discovers that injecting the target token "B" into the prompt increases the probability of that token appearing in the generation via in-context copying mechanisms. This essentially acts as a "soft" few-shot example, biasing the model's output distribution to correct for the dataset skew without necessarily altering the underlying semantic reasoning.

Table 4: **Perplexity on WikiText2.** Language modeling perplexity before and after prepending RESGA- and SAEGA-discovered prompts. The negligible perplexity change across all three model families confirms that our prompts steer persona-relevant behavior without disrupting general language modeling performance.

| Model | Baseline | **RESGA** & **SAEGA** |
|---|---|---|
| Llama | 12.56 | 12.07 |
| Qwen | 10.88 | 12.48 |
| Gemma | 30.51 | 29.19 |

Table 5: **Ablation study results.** We ablate three core components of our framework: the persona steering vector (Random Direction), gradient-guided token mutation (Random Mutation), and the representation-based objective (Task Loss as Objective).

| Method | Sycophancy ↓ | | | Hallucination ↓ | | | Myopic Reward ↓ | | |
|---|---|---|---|---|---|---|---|---|---|
| | Llama | Qwen | Gemma | Llama | Qwen | Gemma | Llama | Qwen | Gemma |
| Random Direction | 74.13 | 84.56 | 81.47 | 68.29 | 54.26 | 84.70 | 50.50 | 48.50 | 61.00 |
| Random Mutation | 68.00 | 71.66 | 76.04 | 50.40 | 45.12 | 54.26 | 52.00 | 46.50 | 55.00 |
| Task Loss as Objective | 55.83 | 73.50 | **58.70** | 46.95 | 45.12 | 54.27 | 48.50 | 48.00 | 51.50 |
| **RESGA** | 49.86 | 50.63 | 62.65 | **45.12** | 41.46 | 51.22 | 38.50 | **38.50** | **34.50** |
| **SAEGA** | **49.84** | **49.95** | 70.70 | 46.95 | **40.85** | **49.14** | **31.50** | 40.50 | 38.50 |

**Semantic Steering via Vocabulary Shift.** In contrast, SAEGA prompts frequently employ *Semantic Steering*, discovering fragments that prime specific reasoning modes rather than simple token forcing. To understand this, we analyzed the shift in output token probabilities induced by the steering prompts ($\Delta\text{Logits} = \text{Logits}_{\text{steered}} - \text{Logits}_{\text{baseline}}$).

- **Sycophancy (Polite Disagreement):** We found that SAEGA systematically suppresses direct agreement markers (`"Yes"`, `"True"`, `"Agree"`) while significantly upweighting semantic pivots such as `"However"` and `"But"`. This suggests the method does not merely force the model to be rude or contradictory, but primes a "critical evaluation" mode where the model creates space for nuance before delivering the factual correction.

- **Myopic Reward (Temporal Priming):** Analysis of the vocabulary shifts showed that RESGA and SAEGA successfully upweighted tokens associated with long-term planning, including `"Future"`, `"Long"`, and `"Wait"`.

### 4.5 Ablation Studies

We validate the three core design choices of our framework through ablation studies (Table 5).

**Persona Steering Vector.** Replacing the persona steering vector with a random direction causes performance to collapse to near or above zero-shot levels across all tasks and models, confirming that a mechanistically grounded steering direction is essential.

**Gradient-Guided Mutation.** Replacing gradient-guided token selection with random token mutation consistently degrades performance, demonstrating that gradient feedback is critical for directing the search toward persona-suppressing prompts.

**Representation-Based Objective.** Replacing the representation projection objective with direct task loss optimization yields unstable results, underperforming RESGA and SAEGA in the majority of settings. This indicates that grounding the objective in model internal representations provides a more reliable optimization signal than task loss alone.

Together, these results confirm that all three components contribute meaningfully to the effectiveness of our framework.

## 5    Conclusion

We present a framework bridging mechanistic interpretability with automatic prompt discovery for persona steering. By constructing persona steering vectors from labeled examples and optimizing prompts via evolutionary gradient ascent, RESGA and SAEGA achieves interpretable control over LLM behavior. Experiments on three persona mitigation tasks demonstrate that discovered prompts achieve significant improvement on sycophancy and myopic reward and consistent improvements on hallucination, substantially outperforming manual prompting and dense steering methods. Mechanistic analysis reveals that SAEGA succeeds by neutralizing rather than shifting behavior, preserving natural activation manifolds, and operating through interpretable feature-level control. Future work could explore semi-supervised approaches to reduce labeled data requirements, investigate cross-model transferability of discovered prompts, and extend to multi-persona steering.

## 6    Limitations

We state our limitations as follows:

- **Labeled Data Requirement.** RESGA and SAEGA require labeled examples exhibiting and lacking the target persona to find steering prompts.

- **Cross-Model Transferability.** The SAE latents and steering vectors are model-specific, requiring retraining for each target model.

- **Fluency-Effectiveness Tradeoff.** Most effective prompts identified by our approach do not show semantic coherence. Future work could aim at using strong linguistic priors to come up with prompts that are effective and human-like at the same time.

## Broader Impact Statement

This work aims to improve LLM safety by suppressing undesired behavioral personas. However, the same pipeline could in principle be adapted to target safety-critical SAE latents, automating jailbreak discovery or eliciting harmful outputs. Deployment should be accompanied by appropriate access controls.

## Acknowledgment

We acknowledge the use of Claude (Anthropic) for manuscript writing assistance and AI-assisted code generation throughout this work.

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

# A   Appendix

This appendix provides extended mechanistic analyses and qualitative examples that support the main claims of the paper but are omitted from the main text due to space constraints.

## A.1   Implementation Details

All experiments were conducted on NVIDIA A100 (40GB) GPUs. Evolutionary Prompt Optimization (EPO) was implemented using the `dreamy` library, and sparse feature extraction was performed using `sae_lens`.

We evaluate Llama-3.1-8B-Instruct, Qwen-2.5-7B-Instruct, and Gemma-3-4B-Instruct. Gated sparse autoencoders were trained on residual stream activations

## A.2   Extended Mechanistic Analysis

We perform a deeper analysis of the internal dynamics of steered models to understand why sparse feature–guided optimization (SAEGA) differs fundamentally from dense residual optimization (RESGA).

### A.2.1   Dynamic Trajectory Analysis (Logit Lens)

To analyze *when* steering takes effect, we apply the Logit Lens, projecting the hidden state at each layer onto the unembedding direction corresponding to the target answers $(\text{Logit}_A - \text{Logit}_B)$.

Figure 7 shows that the baseline model gradually accumulates sycophantic bias, with a sharp increase in later layers. SAEGA maintains near-zero logit differences throughout the network depth, preventing bias accumulation. In contrast, RESGA induces strong mid-layer suppression, suggesting a less stable control mechanism based on over-correction.

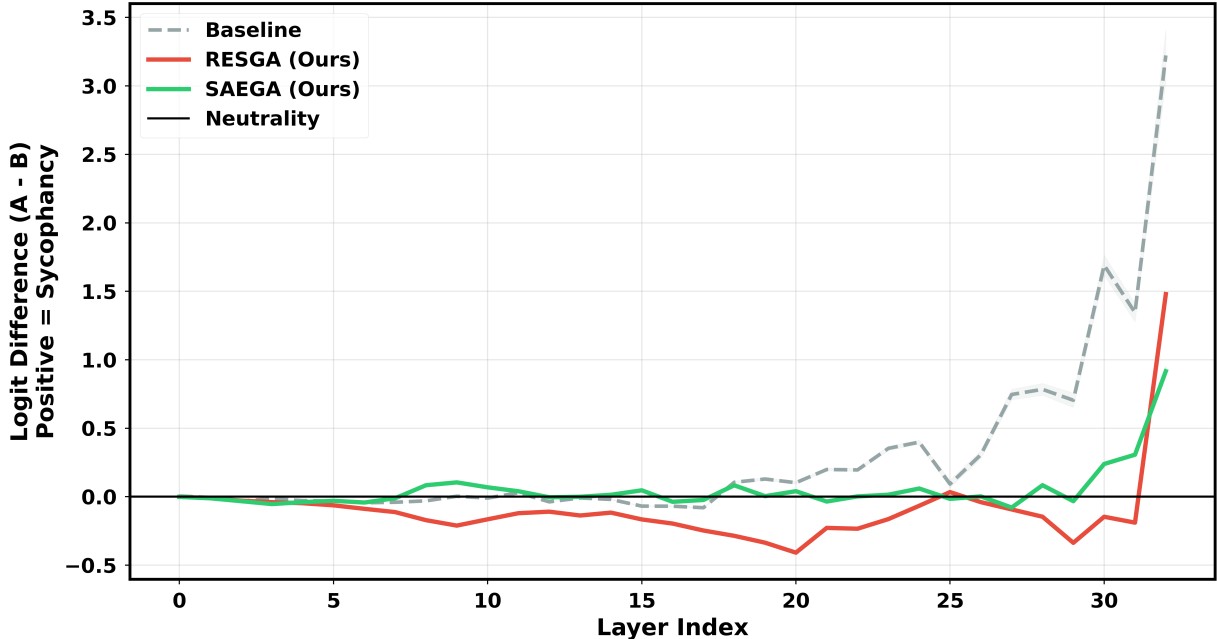

Figure 7: **Logit Lens Analysis.** Trajectory of answer preference $(\text{Logit}_A - \text{Logit}_B)$ across layers. SAEGA maintains neutrality throughout the forward pass, while RESGA induces aggressive mid-layer suppression.

### A.2.2 Layer-wise Steering Mechanics

We compare discrete prompt-based steering (SAEGA, RESGA) with continuous activation steering using dense vectors.

Figure 8 (left) shows that prompt-based methods induce a divergence starting at early layers, allowing the steering signal to compound across depth. In contrast, dense steering vectors act as a localized perturbation at the injection layer.

Figure 8 (right) shows cosine similarity to the baseline trajectory. SAEGA maintains higher similarity than RESGA across intermediate layers, indicating that it preserves more of the model's natural representation geometry.

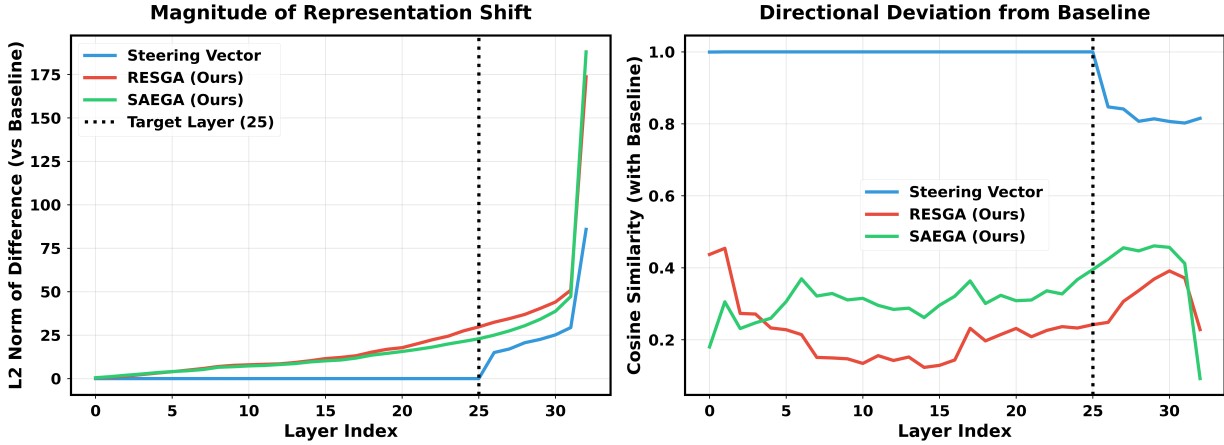

Figure 8: **Layer-wise Steering Mechanics. Left:** L2 distance from baseline across layers. **Right:** Cosine similarity to baseline. SAEGA preserves trajectory structure more effectively than RESGA.

