# OpenReview forum: "Bridging Mechanistic Interpretability and Prompt Engineering with Gradient Ascent for Interpretable Persona Control"
_TMLR — Accepted by TMLR_

### Review · Reviewer_XEcZ · 2026-03-17

**Summary Of Contributions:**

Prompt engineering is standard for controlling the generation process of an LLM.
Which prompt should one use to promote the targeted behavior in LLM?
This difficulty is the main problem that this paper addresses.

Previous work in automatic prompt engineering has largely focused on black-box methods.
While there exist non-black-box methods for this, they are not interpretable---they work in the embedding spaces, not in the natural language space.

This paper introduces RESGA and SAEGA that incrementally update an initial (random) prompt into a prompt that maximally promotes the target behavior in the LLM's output.
First, the algorithm assumes access to two datasets consisting of positive and negative training examples, respectively.
Then, it computes a "persona steering vector" based on those datasets.
The objective is to project the embedding of a prompt into this steering vector and do gradient ascent.
To work in a prompt (discrete) space, an evolutionary algorithm that randomly mutates a token in the prompt based on the gradient is employed.

Experiments show that the proposed methods are better in promoting target behaviors compared to standard baselines.

**Additional Comments:**

1. Improper usage of `\citep` and `\citet` when citing references.
2. Fig. 1 is not vector graphics---pixelated when zoomed in.
3. Improper reference list. No venues are provided. Please use the proper conference/journal/workshop venues. Only use ArXiv if unpublished.

**Audience:**

Yes

**Audience Explanation:**

Automatic prompt engineering is important for steering LLMs' behaviors.

**Claims And Evidence:**

No

**Claims Explanation:**

While the proposed methods are interesting, I have doubts regarding the evaluation:

1. The authors mentioned that there are many lines of work in automatic prompt engineering, including black-box methods (evolutionary algorithms, bandits, etc.). However, they only compare their methods with simple baselines. I suggest adding a broader selection of baselines
2. The methods are complex in that they consist of many different elements from different lines of work, such as fluent gradient ascent, steering vectors, and evolutionary algorithms. However, an ablation study is not currently provided.
3. The main claim of the paper is "interpretability". However, the discovered prompts (Tab. 2) are not really interpretable---they're gibberish. So, I'm not sure how correct the claim is.

**Requested Changes:**

See above.

---

> ### Author Response · Authors · 2026-04-09
> **Reply to the Review by Reviewer XEcZ**
>
> We thank the reviewer for the valuable feedback.
>
> We have addressed the concerns in the revised manuscript as follows :
>
> 1) In Table 1, we have added two black box baselines for comparison -
> i) Greedy Coordinate Gradient (GCG) which optimizes input prompts directly over the output logits.
> ii) ProTeGi, which takes natural language seed prompts as input and then uses an LLM API (we used Gpt-4o) to generate natural language feedback which is used as a “gradient” to then optimize the prompt using bandits and beam search. Results show that the average performance of RESGA and SAEGA is superior to both simple and black box baselines.
>
> 2) We have added an ablation study in Table 5 and Section 4.5 where we ablate the three core components of our algorithms - i) Persona Steering Vectors ii) Gradient based mutation of prompts iii) Representation based objective. The ablation study confirms that each component contributes meaningfully to the effectiveness of our framework.
>
> 3) We agree that the previous framing was not well grounded with the results obtained, we have modified the framing of our claims and removed “natural language” when mentioning the results we have obtained. At the same time, we have now introduced two forms of initialization - i) Random Initialization ii) Seed initialization. For seed initialization, we optimize natural language prompts which are loosely connected to the downstream persona task and we show that we are able to obtain much more interpretable results in terms of human readability. We conclude that the best working prompts obtained are not interpretable in terms of human readability but they do not disrupt general language modeling performance, which we show in Table 4 by calculating perplexity of models on wikitext2 when prepending the prompts obtained by RESGA and SAEGA. We have added Table 3 for seed initialization which shows that we are able to obtain human readable prompts which have competitive performance. For example, for the seed prompt “Let us evaluate this claim purely based on objective evidence.” (score 72.06) , we are able to obtain the prompt “Let's examine the process step by step.” (score 59.35).
>
> 4) Finally, we have redrawn the figure 1 and replaced it in the revised version, and also added some changes to figure 2 to represent both of the initialization modes, and fixed the citations.

---

> > ### Comment · Reviewer_XEcZ · 2026-04-14
> >
> > Thank you for addressing the issues raised by all the reviewers. I do not have further questions.

---

### Review · Reviewer_9fWa · 2026-03-26

**Summary Of Contributions:**

This paper applies evolutionary prompt optimization (EPO) to internal LLM representations to find discrete prompts that suppress behaviors like sycophancy and hallucination. They compare targeting dense residual streams (RESGA) against sparse autoencoder latents (SAEGA).

The core idea, using SAEs as an optimization objective for discrete prompt search, is practical. The mechanistic analysis is the highlight here; tracking L0 sparsity effectively demonstrates why SAE-guided search avoids the representation collapse typically caused by dense activation steering.

**Audience:**

Yes

**Audience Explanation:**

Bridging mechanistic interpretability (SAEs) with discrete prompt optimization is an active area in LLM alignment. The analytical insights regarding how SAE-guided search better preserves the activation manifold will be of broad interest to the TMLR community.

**Broader Impact Concerns:**

The same gradient ascent pipline used here to suppress bad personas could reasonably be adapted to target toxic SAE latents instead, raising concerns around jailbreak automation. The authors should acknowledge and discuss this dual-use risk.

**Claims And Evidence:**

No

**Claims Explanation:**

There is a disconnect between the paper's narrative framing and its empirical results.

First, the abstract and introduction heavily market the discovery of "interpretable natural language prompts" via "fluent gradient ascent." Yet Table 2 shows the outputs are adversarial token sequences. A cross-entropy penalty in the objective does not, on its own, make token fragments fluent, and framing the results as "natural language" may overstate what the method achieves.

Second, the authors claim successful "persona control" but evaluate this entirely using conditional A/B log-probabilities. Since the injected prompts are heavily off-manifold, prepending them may compromise the LLM's basic ability to generate coherent text. Under those conditions, a shift in the log-prob ratio is hard to read as clean evidence of successful steering.

**Requested Changes:**

1. The framing around "fluent" or "natural language" prompts would benefit from some adjustment. Rather than marketing the outputs as natural language, the abstract and intro could more accurately describe them as mathematically constrained discrete tokens, with interpretability attributed to the targeted SAE latents themselves.

2. A brief generative sanity check would help address a potential concern about the evaluation. Even something lightweight, like perplexity on WikiText or a few qualitative generation samples, would reassure readers that the SAEGA prompt does not degrade the model's ability to produce coherent English.

3. It would be helpful to include a Broader Impact Statement given the dual-use nature of the method.

4. The paper would also benefit from a comparison against a standard discrete optimization baseline like GCG, which optimizes output logits directly. This would make it easier to see what the SAE-latent objective contributes beyond what a simpler black-box approach could achieve.

---

> ### Author Response · Authors · 2026-04-09
> **Reply to the Review by Reviewer 9fWa**
>
> We thank the reviewer for the valuable feedback and are glad that the reviewer highlighted the result where we expose the fundamental limitation of dense steering vectors that they cause distributional shifts that are not induced by prompt based methods like ours. In the revised version, we have added more emphasis to this result in the introduction.
>
> We have addressed the concerns in the revised manuscript as follows :
>
> 1) We agree that the previous framing was not well grounded with the results obtained, we   have modified the framing of our claims and removed “natural language” when mentioning the results we have obtained. At the same time, we have now introduced two forms of initialization - i) Random Initialization ii) Seed initialization. For seed initialization, we optimize natural language prompts which are loosely connected to the downstream persona task and we show that we are able to obtain much more interpretable results in terms of human readability. We conclude that the best working prompts obtained are not interpretable in terms of human readability but they do not disrupt general language modeling performance, which we show in Table 4 by calculating perplexity of models on wikitext2 when prepending the prompts obtained by RESGA and SAEGA. We have added Table 3 for seed initialization which shows that we are able to obtain human readable prompts which have competitive performance. For example, for the seed prompt “Let us evaluate this claim purely based on objective evidence.” (score 72.06) , we are able to obtain the prompt “Let's examine the process step by step.” (score 59.35).
>
> 2) We agree to the reviewer’s concern regarding the dual-use risk and we have added a broader impact statement to the revised manuscript.
>
> 3) In Table 1, we have added two black box baselines for comparison -
> i) Greedy Coordinate Gradient (GCG) which optimizes input prompts directly over the output logits.
> ii) ProTeGi, which takes natural language seed prompts as input and then uses an LLM API (we used Gpt-4o) to generate natural language feedback which is used as a “gradient” to then optimize the prompt using bandits and beam search.
>
> 4) We have added an ablation study in Table 5 and Section 4.5 where we ablate the three core components of our algorithms - i) Persona Steering Vectors ii) Gradient based mutation of prompts iii) Representation based objective. The ablation study confirms that each component contributes meaningfully to the effectiveness of our framework.
>
> 5) We have also replaced figure 1 owing to AI generation concerns noted by other reviewers and we have made some changes to figure 2 to represent both initialization modes.

---

### Review · Reviewer_cote · 2026-04-03

**Summary Of Contributions:**

- They introduce a framework that adapts gradient ascent to systematically discover natural language prompts for targeted persona control in Large Language Models (LLMs). This approach successfully bridges the gap between mechanistic interpretability and automated prompt engineering.

- They propose two distinct methods for constructing the underlying persona steering vectors. RESGA builds steering vectors by computing the direct, dense mean differences in residual stream representations. SAEGA constructs steering vectors by isolating causally relevant, interpretable features using pretrained SAE latents.

**Audience:**

Yes

**Audience Explanation:**

The core findings of this paper would definitely interest specific sub-communities within the TMLR audience.

**Broader Impact Concerns:**

No ethical concerns.

**Claims And Evidence:**

No

**Claims Explanation:**

The authors repeatedly claim their framework discovers "natural language prompts" and utilizes "fluent gradient ascent" to control the fluency of the steering prompts. However, their own evidence entirely contradicts this. Table 2 shows that the discovered prompts are complete gibberish, featuring fragmented syntax, random Unicode characters, and mixed language. The authors even admit in their limitations that "Most effective prompts identified by our approach do not show semantic coherence". Claiming a "fluent" and "natural language" methodology while producing unreadable output is a significant misrepresentation of the results.

**Requested Changes:**

- Figures 1 contain visible artifacts of AI image generation, including garbled, hallucinated text such as "Graidual Accent," "Finidual Vect," "Predse Pesona Stuend," and "Empathetis". I recommend this figure should be entirely redrawn and proofread to meet academic publication standards. The presence of unvetted AI-generated diagrams severely undermines the perceived rigor of the surrounding empirical analysis.

- The authors state that an error rate of ~50% on the sycophancy task corresponds to "perfect neutralization", where the model neither systematically agrees nor disagrees. However, because the injected prompts are effectively gibberish, a 50% error rate on a binary classification task could equally indicate that the model's fundamental reasoning capabilities have been broken, reducing it to random guessing. The authors must provide evaluation metrics on standard capability benchmarks (e.g., MMLU, GSM8K) or report generation perplexity while the discovered prompts are active to prove that the model retains its general competence.

- The abstract and conclusion claim the method offers a "new paradigm for controllable and interpretable behavior modification". While the selection of SAE features provides mechanistic grounding , the ultimate intervention is fundamentally uninterpretable to human operators. The framing must be adjusted to clearly separate the interpretability of the vector construction from the opaqueness of the resulting prompt interface.

---

> ### Author Response · Authors · 2026-04-09
> **Reply to the Review by Reviewer cote**
>
> We thank the reviewer for the valuable feedback.
>
> We have addressed the concerns in the revised manuscript as follows :
>
> 1) We have completely replaced figure 1.
> 2) We agree that the previous framing was not well grounded with the results obtained, we have modified the framing of our claims and removed “natural language” when mentioning the results we have obtained. At the same time, we have now introduced two forms of initialization - i) Random Initialization ii) Seed initialization. For seed initialization, we optimize natural language prompts which are loosely connected to the downstream persona task and we show that we are able to obtain much more interpretable results in terms of human readability. We conclude that the best working prompts obtained are not interpretable in terms of human readability but they do not disrupt general language modeling performance, which we show in Table 4 by calculating perplexity of models on wikitext2 when prepending the prompts obtained by RESGA and SAEGA. We have added Table 3 for seed initialization which shows that we are able to obtain human readable prompts which have competitive performance. For example, for the seed prompt “Let us evaluate this claim purely based on objective evidence.” (score 72.06) , we are able to obtain the prompt “Let's examine the process step by step.” (score 59.35). We have added changes to figure 2 to represent both of the initialization modes.
> 3) We have added an ablation study in Table 5 and Section 4.5 where we ablate the three core components of our algorithms - i) Persona Steering Vectors ii) Gradient based mutation of prompts iii) Representation based objective. The ablation study confirms that each component contributes meaningfully to the effectiveness of our framework.
> 4) In Table 1, we have added two black box baselines for comparison -
> i) Greedy Coordinate Gradient (GCG) which optimizes input prompts directly over the output logits.
> ii) ProTeGi, which takes natural language seed prompts as input and then uses an LLM API (we used Gpt-4o) to generate natural language feedback which is used as a “gradient” to then optimize the prompt using bandits and beam search.

---

### Decision · Action_Editor_qAj4 · 2026-06-21

**Recommendation:** Accept as is

**Audience:**

Yes

**Audience Explanation:**

The intersection of mechanistic interpretability (specifically using Sparse Autoencoders) and automated discrete prompt engineering is an active, highly relevant area in LLM alignment and safety. It's clear that the paper is in-scope and interesting for TMLR audience.

**Claims And Evidence:**

Yes

**Claims Explanation:**

Initially, reviewers raised concerns regarding a disconnect between the paper's claims of discovering "fluent natural language" prompts and the empirical results (which showed unreadable, adversarial tokens). Reviewers also questioned whether the method genuinely steered persona behavior or simply degraded the model's foundational reasoning capabilities. In the revision, the authors addressed these flaws. They recalibrated their claims, introduced a "Seed Initialization" mode that successfully produces human-readable prompts, and provided a generative sanity check demonstrating that the injected prompts do not disrupt general language modeling performance. Furthermore, the inclusion of appropriate standard baselines (GCG and ProTeGi) and a comprehensive ablation study solidifies the empirical evidence supporting the proposed RESGA and SAEGA methods. The AE believes that this criterion is therefore met at the end of the discussion period.